# Atypical Chemokine Receptor CCRL2 Shapes Tumor Spheroid Structure and Immune Signaling in Melanoma

**DOI:** 10.3390/biom15081150

**Published:** 2025-08-11

**Authors:** Diana Al Delbany, Mai Chi Duong, Marius Regin, Arkajyoti Sarkar, Ayoub Radi, Anne Lefort, Frédérick Libert, Marc Parmentier, Claudia Spits

**Affiliations:** 1Research Group Genetics, Reproduction and Development, Faculty of Medicine and Pharmacy, Brussels Health Campus, Vrije Universiteit Brussel (VUB), Laarbeeklaan 103, 1090 Brussels, Belgium; chi.mai.duong@vub.be (M.C.D.); marius.regin@vub.be (M.R.); claudia.spits@vub.be (C.S.); 2Department of Biochemistry, Military Hospital 175, 786 Nguyen Kiem Street, Ho Chi Minh City 71409, Vietnam; 3Liver Cell Biology Research Group, Vrije Universiteit Brussel, 1090 Brussels, Belgium; arkajyoti.sarkar@vub.be; 4Institute of Jacques-Dumont Interdisciplinary Research in Molecular Human Biology (Jacques-Dumont IRIBHM), Université Libre de Bruxelles, 1070 Brussels, Belgiumanne.lefort@ulb.be (A.L.); frederick.libert@ulb.be (F.L.); marc.parmentier@ulb.be (M.P.); 5BRIGHTcore Facility, 1070 Brussels, Belgium

**Keywords:** CCRL2, atypical chemokine receptor, tumor spheroids, melanoma

## Abstract

C-C motif chemokine receptor-like 2 (CCRL2) is an atypical chemokine receptor (ACKR) that binds chemerin with high affinity but lacks classical G protein-coupled signaling. Instead, it functions as a non-signaling presenter of chemerin to CMKLR1-expressing cells, modulating antitumor immunity. CCRL2 is highly expressed in the tumor microenvironment and various human cancers, and its expression has been linked to delayed tumor growth in mouse models, primarily through the chemerin/CMKLR1 axis. While CCRL2’s role in immune surveillance is well established, its tumor cell-intrinsic functions remain less clear. Here, we investigated the impact of CCRL2 overexpression and knockout on tumor cell behavior in vitro. Although CCRL2 did not affect proliferation, migration, or clonogenicity in B16F0 melanoma and LLC cells, it significantly influenced spheroid morphology in B16F0 cells. Transcriptomic analysis revealed that CCRL2 modulates innate immune signaling pathways, including TLR4 and IFN-γ/STAT1, with context-dependent downstream effects. These findings suggest that CCRL2 shapes tumor architecture by rewiring inflammatory signaling networks in a cell-intrinsic manner. Further studies in other cancer types and cell models are needed to determine whether CCRL2’s regulatory role is broadly conserved and to explore its potential as a therapeutic target in solid tumors.

## 1. Introduction

CCRL2 is a seven-transmembrane G protein-coupled receptor (GPCR), also known as human chemokine receptor (HCR) or chemokine receptor on activated macrophages (CRAM) in humans, and as lipopolysaccharide-inducible CC-chemokine receptor (L-CCR) in mice. The *CCRL2* gene is located within the main cluster of genes encoding receptors for inflammatory CC-chemokines, sharing up to 40% sequence identity with members of this family, including *CCR1*, *CCR2*, *CCR3*, and *CCR5* [1,2,3,4,5]. The CC-chemokine receptor family possesses a highly conserved sequence in the second intracellular loop, known as the conical DRYLAIV motif, which is essential for conventional signaling through G proteins. This motif is significantly modified in human CCRL2 (QRYLVFL) and mouse Ccrl2 (QGYRVFS), preventing CCRL2 from coupling with Gi proteins and triggering classical chemokine signaling or cell migration. These characteristics link CCRL2 to the family of ACKRs [6]. Moreover, chemokine receptors, including ACKRs, play pivotal roles in tumor progression by regulating cell proliferation, adhesion, and migration, either through autocrine signaling or through crosstalk with other membrane receptors [7,8].

CCRL2 binds chemerin, a multifunctional protein with well-established functions in inflammation, cancer, adipogenesis, and glucose homeostasis, with high affinity without inducing receptor internalization or activating downstream signaling cascades [4,9,10,11,12]. Instead, CCRL2 concentrates chemerin on the cell surface and presents it to other cells expressing the main functional chemerin receptor, CMKLR1 [13,14]. Therefore, CCRL2 functions as non-signaling receptor and chemokine presenter, giving it a unique functional profile among non-signaling GPCRs [4,13,15]. CCRL2 has garnered attention for its regulatory functions in inflammation and cancer. CCRL2 is expressed in various leukocyte subsets and barrier cells, with its expression upregulated by inflammatory stimuli such as LPS, IFN-γ, TNF-α, and GM-CSF [5]. Several studies have extensively investigated the role of CCRL2 in regulating inflammatory responses across various pathological conditions, including hypersensitivity reactions, arthritis, and experimental autoimmune encephalomyelitis [5,15,16,17,18,19,20].

CCRL2 expression has been reported across a range of malignancies and tumor-derived cell lines, including human glioblastoma [21], colorectal carcinoma [22], breast cancer [23], prostate cancer [24], and non-small-cell lung cancer [20,25,26]. In metastatic melanoma, CCRL2 expression has been shown to significantly correlate with overall patient survival [27]. More recently, CCRL2 was found to be overexpressed in acute myeloid leukemia (AML) harboring TP53 mutations [28]. CCRL2 exhibits context-dependent roles in cancer, acting as a tumor suppressor in some contexts while promoting tumor progression in others.

In a previous study, we demonstrated that the loss of Ccrl2 in the host accelerates papilloma development in a chemical model of skin carcinogenesis, whereas CCRL2 overexpression in B16 melanoma or LLC cells delays tumor graft growth in vivo [29]. These effects are mediated by the chemerin/CMKLR1 axis, with CCRL2 enhancing chemerin bioavailability by binding and concentrating its bioactive form. This, in turn, activates CMKLR1-expressing cells and suppresses tumor neoangiogenesis, recapitulating the phenotype observed in chemerin-expressing tumors. Notably, the phenotypes associated with CCRL2 gain or loss of function were significantly reduced, though not completely abrogated, in chemerin- or CMKLR1-deficient models, indicating that additional, chemerin-independent mechanisms may be involved. Furthermore, CCRL2 expression did not affect B16 or LLC cell proliferation in vitro. Together, these findings suggest that CCRL2 may influence tumorigenesis through mechanisms beyond chemerin signaling and independent of direct effects on cell growth [29]. To investigate this further, the present study explores whether CCRL2 influences other tumor cell-intrinsic properties, including migration, clonogenicity, and 3D spheroid formation.

## 2. Materials and Methods

### 2.1. Cell Lines

Murine B16-F0 melanoma and Lewis lung carcinoma (LLC) cell lines were purchased from the American Type Culture Collection (ATCC). Cells were cultured in RPMI-1640 medium (Life Technologies, Merelbeke, Belgium) supplemented with 10% fetal bovine serum (FBS-Gibco, Paisley, UK), 1% sodium pyruvate (Gibco, Paisley, UK), 100 U/mL penicillin, and 100 µg/mL streptomycin (Invitrogen, Carlsbad, CA, USA). Cultures were maintained at 37 °C in a humidified incubator with 5% CO_2_. Mycoplasma contamination was regularly monitored, and all cell lines were confirmed negative.

### 2.2. Generation of Ccrl2-Knockout and CCRL2-Overexpression Cell Lines

LLC and B16 clones with *Ccrl2*-knockout or CCRL2-overexpression were generated and characterized as described in our previous work [29].

The *Ccrl2* gene was knocked out using a CRISPR/Cas9n system with paired single-guide RNAs (sgRNAs) targeting the region immediately downstream of the AUG start codon in the second and only coding exon of *Ccrl2*. The designed sgRNAs were cloned into a GFP-tagged Cas9n expression vector and co-transfected into LLC and B16 cells using Lipofectamine 2000 transfection reagent (ThermoFisher, Waltham, MA, USA) according to the manufacturer’s instructions. GFP-positive single cells were sorted by flow cytometry, and at least five clones per cell line were expanded. Clones were screened by PCR (5′-TGTCGGATGGAGGGGAATCA-3′ as the forward primer and 5′-CCAAGATAAACACCGCCAGC-3′ as the reverse primer). DNA heteroduplex mobility assay (HMA) on polyacrylamide gels was used to detect small deletions or insertions (1 to 4 bp) using 5′-TGTCGGATGGAGGGGAATCA-3′ as the HMA-forward primer and 5′-CCAAGATAAACACCGCCAGC-3′ as the HMA-reverse primer. The presence of the frameshift mutation and premature stop codon in each clone was confirmed by Sanger sequencing. Two independent clones from both B16 and LLC cell lines (B16-CCRL2^KO^ and LLC- CCRL2^KO^) were selected for further analysis.

For CCRL2 overexpression, LLC and B16 cells were stably transfected with a bicistronic plasmid (pCDNeo) encoding a codon-optimized Ccrl2 cDNA and a neomycin resistance gene under the EF-1α promoter. Cells were seeded in serum-free RPMI-1640 medium and transfected using the X-tremeGENE Transfection reagent (Roche, Mannheim, Germany) according to the manufacturer’s instructions. After 48 h, cells were re-seeded in a medium containing 800 μg/mL neomycin (G1418-Invivogen, Toulouse, France), and individual resistant clones were isolated. Expression of *Ccrl2* was confirmed by PCR using primers specific for the codon-optimized *Ccrl2* (5′-ACGAGCCCAGAATGGAGAGA-3′ as the forward primer and 5′-GCTTGTGCAGGTCGTACTGT-3′ as the reverse primer). Protein expression was validated by flow cytometry using a PE-conjugated anti-mouse CCRL2 antibody (clone BZ2E3, BD Pharmingen, San Diego, CA, USA). Two independent clones from both B16 and LLC cell lines (B16-CCRL2^OE^ and LLC- CCRL2^OE^) were selected for further analysis.

### 2.3. Cell Migration Assay

B16 and LLC cell lines were seeded in Culture-Insert 2 wells (Ibidi GmbH, Gräfelfing, Germany, Cat.No:81176) at a density of 14,000 cells per insert in RPMI medium and cultured to confluence. Following cellular attachment, the inserts were carefully removed to create a defined 500 μm cell-free gap. Migration was assessed in serum-free media throughout the experiment to minimize cell proliferation. Phase-contrast images were acquired at multiple time points (0–24 h for B16, 0–48 h for LLC), using a ZOE Fluorescent Cell Imager (BioRad Laboratories, Hercules, CA, USA) at 20× magnification. Migration was quantified by measuring the percentage of the remaining cell-free area over time in three independent fields per wound, using ImageJ software (version 1.53c, National Institutes of Health, Bethesda, MD, USA).

### 2.4. Cell Colony Formation Assay

Cells were seeded at a density of 500 cells per well in six-well plates and cultured in RPMI medium supplemented with 10% FBS at 37 °C. After one week, colonies were fixed with ice-cold 100% methanol for 10 min on ice, stained with 0.5% crystal violet (Sigma-Aldrich, sourced from Merck NV/SA, Overijse, Belgium; Cat. #C6158) for two hours at room temperature. Plates were rinsed thoroughly with distilled water and air-dried. Colony images were acquired using a digital camera. Quantification was performed using ImageJ software. Colonies were defined as clusters consisting of at least 50 cells. Thresholded overlays were applied to ensure accurate and consistent identification of colonies. Image analysis was performed as previously described [30]. Clearly delineated colonies were counted individually, while adjacent clusters that likely originated from a single colony were counted as one. In cases where colony boundaries were ambiguous, quantification was considered uncertain. Wells with overgrown cells, where discrete colony identification was not feasible, were excluded from the analysis.

### 2.5. Spheroid Formation Assay

Three-dimensional (3D) spheroid culture was performed using the liquid overlay technique, as previously described [31,32], with minor adjustments. Prior to spheroid assembly, agarose (Invitrogen) was dissolved in distilled water at a concentration of 0.01 g/mL by heating it to boiling (~80 °C). A total of 150 µL of the agarose solution was added to each well of a 48-well plate to prevent attachment of the tumor cells to the plastic surface. The agarose was allowed to polymerize at 4 °C for 10 min and at room temperature for 20 min to ensure complete solidification. B16 and LLC tumor cells were harvested using trypsin, and the cell count was determined using the EVE automatic cell counter (NanoEnTek Inc., Seoul, South Korea). A total of 1000 cells were plated per well on top of the solidified agarose substrate in 300 µL of RPMI medium, supplemented with 1%, 5%, or 10% FBS, with or without 2 mg/mL methylcellulose (MC). Spheroid formation was allowed to proceed for 3–4 days at 37 °C in a 5% CO_2_ incubator. After spheroid formation, the medium was refreshed every 2 days by removing 100 µL and replacing it with an equal volume of fresh complete growth medium. Spheroids were monitored daily and imaged on days 3, 6, and 9 using a ZOE Fluorescent Cell Imager (BioRad Laboratories, Hercules, CA, USA) at 20× magnification. Quantification of spheroid area, perimeter, diameter, and circularity was performed using ImageJ with automated batch processing through a custom script available at https://github.com/arka2696/HPC-Organoid-Quant.git (accessed on 25 July 2025).

### 2.6. Flow Cytometry Analysis

For flow cytometry, multiple spheroids (n = 6–10) were collected and pooled into a single sample to ensure sufficient cell yield for antibody staining. Spheroids were transferred to a 15 mL conical tube using wide-bore pipette tips to minimize mechanical shear stress. Samples were washed once with PBS without Ca^2+^/Mg^2+^ to remove residual culture medium. Dissociation was performed by incubating the spheroids in PBS supplemented with 5 mM EDTA (Thermo Fisher Scientific, Waltham, MA, USA) at 37 °C for 15–20 min. During incubation, spheroids were gently triturated every 5 min using a P1000 pipette to facilitate dissociation into single cells. Following enzymatic digestion, the reaction was blocked by adding an equal volume of FACS buffer (PBS containing 1% FCS, 1 mM EDTA and 0.1% NaN3). The cell suspension was passed through a 40-µm cell strainer (Falcon) to remove aggregates. Single-cell suspensions were then stained for 30 min at 4 °C with anti-CCRL2 (clone BZ2E3, BD Pharmingen, 564946), and anti-E-cadherin (clone ECCD-2, ThermoFisher Scientific, 13-1900). Flow cytometry analysis was performed on LSRFortessa instrument ((BD Biosciences, San Jose, CA, USA) and analyzed using the lowJo software (version 10.10.0, BD Biosciences, San Jose, CA, USA).

### 2.7. RNA Sequencing and Transcriptome Analysis

Tumoral spheroids were lysed in RLT lysis buffer (Qiagen, Hilden, Germany), to which we added 0.01% β-mercaptoethanol (Sigma). RNA was extracted with the RNAeasy Minikit (Qiagen) following the manufacturer’s instructions. RNA quality was checked using a Fragment Analyzer (Agilent Technologies, Santa Clara, CA, USA). Indexed cDNA libraries were obtained using the NuGEN SoLo RNA sample preparation kit (Tecan Group Ltd., Männedorf, Switzerland) following the manufacturer’s recommendations. The multiplexed libraries (10 pM) were loaded on flow cells, and sequences were produced using a HiSeq PE Cluster Kit v4 and TruSeq SBS Kit v3-HS from a NovaSeq 6000 system (Illumina, San Diego, CA, USA) at the Brussels Interuniversity Genomics High Throughput core (www.brightcore.be). Approximately 29 to 33 million paired-end reads per sample were mapped against the mouse reference genome (GRCm38.p4/mm10) using the STAR software (STAR v2.7.3a) to generate read alignments for each sample. The annotation file Mus_musculus.GRCm38.90.gtf was obtained from the ftp.Ensembl.org server. Following transcript assembly, gene-level counts were obtained using HTSeq.

Count matrices were imported into R (version 4.3.1) for further processing. Differentially expressed genes (DEGs) between groups were identified using the DESeq2 package. Genes with a total raw count of at least 10 across all samples were included in the downstream analysis. Genes were considered significantly differentially expressed if they had a log2 fold change (log2FC) greater than 1 or less than −1, and a false discovery rate (FDR)-adjusted *p*-value < 0.05.

Volcano plots of DEGs were generated using the ggplot2 package in R. Principal component analysis (PCA) and hierarchical clustering were performed using variance-stabilizing transformed (VST) counts. The heatmaps were created with the ComplexHeatmap package, while PCA was conducted using the prcomp function and visualized with ggplot2.

Upstream pathway analysis was performed using Ingenuity Pathway Analysis (IPA) ([33]; QIAGEN Inc., https://www.qiagenbioinformatics.com/products/ingenuity-pathway-analysis, accessed on 25 July 2025). For further analysis, we included only upstream regulators with activation Z-scores greater than 2 or less than −2 and adjusted *p*-values < 0.05.

### 2.8. Statistical Analysis

Statistical analyses and data visualization were performed using Prism 6 and InStat (GraphPad Software, version 10.2.3 (403), San Diego, CA, USA). Statistical significance was assessed using one-way or two-way ANOVA, followed by Tukey’s or Dunnett’s multiple comparisons tests, respectively. *p*-values less than 0.05 were considered statistically significant.

## 3. Results

### 3.1. CCRL2 Expression Does Not Significantly Alter Tumor Cell Migration in B16 or LLC Cells

Migration is a fundamental process in immune surveillance, enabling leukocytes to navigate toward sites of inflammation, infection, or malignancy. Cell migration and invasion may widely vary between different cell types. CCRL2 was shown to influence dendritic cell migration in the lung [16]. As tumor cell migration shares many similarities with leukocyte trafficking, we investigated the role of CCRL2 in tumor cell motility. Migration capacity was assessed in CCRL2-overexpressing (B16-CCRL2^OE^ and LLC-CCRL2^OE^) and CCRL2-knockout (B16-CCRL2^KO^ and LLC-CCRL2^KO^) variants of B16 and LLC cells, in comparison to their respective parental lines.

In the B16 melanoma model, minimal migration was observed across all cell lines at 4, 6, and 8 h following insert removal. By 14, 18, and 24 h, wound closure reached approximately 32%, 46%, and 60%, respectively, for both B16 and B16-CCRL2^KO^ cells, while B16-CCRL2^OE^ cells achieved 23%, 38.6%, and 50% coverage (Figure 1A,B). Although CCRL2 overexpression was associated with a slight reduction in migration rates, the differences were not statistically significant (*p* > 0.05). In the LLC lung carcinoma model, the overall migratory activity was reduced relative to B16 cells (Figure 1C,D), with no significant differences observed among parental, CCRL2-overexpressing, and CCRL2-deficient lines at 14, 18, 24, or 48 h post-insert removal (*p* > 0.05) (Figure 1C,D). These results indicate that CCRL2 expression does not significantly modulate the in vitro migratory capacity of B16 or LLC tumor cells.

### 3.2. Changes in CCRL2 Expression Do Not Influence Clonogenic Efficiency but Modulate Colony Structure

Previous studies have demonstrated that CCRL2 expression influences the clonogenic potential of various cancer cell lines. In human breast cancer cells (MDA-MB-231), overexpression of CCRL2 significantly inhibited colony formation [34]. Moreover, the knockdown of CCRL2 affected the clonogenic capacity of different human colorectal cancer cell lines (SW620, Caco2, and LS174T), with a reduction in colony formation for LS174T cells and a slight increase for SW620 and Caco2 cells [22]. These findings suggest that CCRL2’s role in regulating tumor cell clonogenicity is context-dependent and may vary across different cancer types. To evaluate the role of CCRL2 in tumor cell survival and clonogenic potential, we performed colony formation assays using B16 melanoma and LLC cell lines.

Under the assay conditions, LLC cells, regardless of CCRL2 expression, did not form stable colonies, likely due to their sensitivity to mechanical disruption during processing, consistent with their mesenchymal-like phenotype.

B16 cells formed colonies across all groups, with no significant differences in colony number (Figure 2A,B). However, differences in colony morphology and size were observed. B16-CCRL2^KO^ cells formed significantly larger and more dispersed colonies compared to both control and CCRL2^OE^ cells (Figure 2C and Appendix A). Additionally, B16-CCRL2^OE^ cells tended to form smaller and more compact colonies (Figure 2C and Appendix A). These results were confirmed in an independent B16 clone overexpressing or knockout for CCRL2, with no significant difference in the outcome (Appendix A). This suggests that while CCRL2 expression does not affect the colony-forming capacity of B16 cells, its loss may influence cell–cell or cell–matrix interactions, thereby altering colony architecture and size.

### 3.3. CCRL2 Expression Impacts Spheroid Morphology in B16 but Not LLC Cells

Our results suggest that CCRL2 expression may influence cell–cell adhesion, thereby potentially contributing to its role in regulating tumor growth *in vivo*, as we demonstrated previously in graft models [29]. To further investigate this, we used tumor spheroids as a more physiologically relevant in vitro model that better mimics the spatial organization and molecular features of solid tumors [35,36,37], providing a robust system to evaluate the effects of CCRL2 expression on tumor cell behavior, three-dimensional growth, and organization.

In LLC cells, CCRL2 expression had no apparent effect on spheroid morphology or growth. All variants formed round, compact spheroids with well-defined boundaries (Figure 3A, left; Appendix A), and no significant differences in size or shape were observed (Figure 4A and Appendix A).

In contrast, B16 melanoma cells exhibited distinct spheroid phenotypes depending on CCRL2 expression (Figure 3A, right; Appendix A). B16-CCRL2^OE^ cells consistently formed compact, mass-like spheroids, whereas control B16 cells formed loosely assembled structures with irregular margins and larger areas. B16-CCRL2^KO^ cells generated large, disorganized, and grape-like spheroids with loose cell–cell contacts (Figure 4A and Appendix A). Similar results were obtained using an independent B16 clone with CCRL2 overexpression or knockout, confirming the reproducibility of the findings (Appendix A).

These results suggest that CCRL2 has a cell-type dependent effect, in our setting specifically modulating the spheroid-forming ability of B16 melanoma cells. Because tumor spheroid formation is sensitive to serum levels and medium composition [38], we tested low-serum conditions and methylcellulose (MC) supplementation to minimize external variability and support anchorage-independent growth [39], with the aim of confirming that the observed effects in B16 cells were specifically attributable to CCRL2 expression, and not confounded by culture conditions. Under 1% FBS, control B16 spheroids appeared more compact and rounded throughout the experiment (Figure 3B, Figure 4B and Appendix A). In both 5% FBS without MC and 10% FBS with MC (Appendix A), control spheroids maintained a round shape until day 3 (Figure 4C,D); however, they were more compact under low-serum conditions (Appendix A). After day 3, spheroids in these conditions began to lose their round morphology and became increasingly irregular over time, resembling the disorganized spheroids observed in 10% FBS without MC (Figure 3A). The addition of MC to 5% FBS moderately improved spheroid roundness and structural integrity until day 6 (Figure 3E), but this effect was lost by day 9 (Appendix A). This suggests that both reduced serum levels and anchorage-independent conditions promote more organized spheroid formation in control B16 cells, potentially by limiting serum-driven proliferation or spreading.

Notably, B16-CCRL2^OE^ spheroids displayed minimal morphological changes across all tested conditions consistently maintaining a compact, well-organized structure and smaller overall size (Figure 3, Figure 4, Appendix A). Moreover, B16-CCRL2^KO^ spheroids remained large (Figure 4 and Appendix A), disorganized, and loosely aggregated regardless of serum or MC levels (Figure 3 and Appendix A), reinforcing the role of CCRL2 in maintaining spheroid integrity.

Together, these findings demonstrate that CCRL2 expression strongly influences the structural organization of B16 melanoma spheroids, independent of external culture conditions. Notably, these experiments were conducted in the absence of chemerin and CMKLR1, indicating that the observed effects were independent of the chemerin/CMKLR1 axis and solely attributable to CCRL2 expression in tumor cells.

### 3.4. CCRL2 Expression Correlates with E-Cadherin Levels in B16 Spheroids

Spheroid formation and maintenance involve dynamic cell–cell and cell–extracellular matrix (ECM) interactions that progress through three distinct phases: initial cell aggregation, spheroid compaction, and subsequent growth [40]. The early aggregation phase is primarily driven by integrin–ECM interactions, whereas the compaction phase depends largely on cadherin-mediated cell–cell adhesion, particularly via E-cadherin. E-cadherin (CDH1) plays a pivotal role in maintaining epithelial integrity by mediating strong intercellular adhesion [41]. Its expression is essential for the structural organization of spheroids in various cellular systems. In breast cancer models, overexpression of CCRL2 has been shown to inhibit CCL2-induced p38 MAPK phosphorylation, resulting in restored E-cadherin expression and reduced cellular invasiveness [34]. Given this, we analyzed the expression of E-cadherin and CCRL2 in day 6 spheroids formed from B16 and LLC cells using flow cytometry. In both control B16 and B16-CCRL2^KO^ spheroids, neither CCRL2 nor E-cadherin were detected at the cell surface (Figure 5A–C). In contrast, B16 spheroids overexpressing CCRL2 exhibited a clear induction of E-cadherin, which correlated with their more compact and organized morphology (Figure 5A–C). Conversely, none of the LLC spheroid variants, including those overexpressing CCRL2, showed surface expression of E-cadherin (Figure 5D–F). These findings suggest that B16 cells, but not LLC cells, are responsive to CCRL2-mediated signaling leading to E-cadherin induction under spheroid culture conditions. To determine whether this regulation is specific to the 3D context, we analyzed the same cell lines cultured under 2D conditions. In 2D culture, E-cadherin was not expressed in any cell line or condition, whereas CCRL2 was detected only in the CCRL2^OE^ populations (Appendix A). Together, these results indicate that CCRL2 overexpression induces E-cadherin expression specifically in the 3D spheroid environment, and that this effect is cell type-dependent.

### 3.5. RNA Sequencing Reveals CCRL2-Dependent Transcriptional Programs in B16 Tumor Spheroids

To elucidate the molecular mechanisms underlying the phenotypic differences observed in B16 melanoma spheroids, we performed bulk RNA sequencing of B16, CCRL2-overexpressing, and CCRL2-knockout spheroids. Spheroids were cultured in 10% FBS-containing medium and harvested on day 6, corresponding to the time point when morphological differences were most evident (Figure 3A). Principal component analysis (PCA) revealed clear separation of the three groups, highlighting global transcriptional divergence linked to CCRL2 expression levels (Figure 6A). Differential gene expression analysis identified 553 genes upregulated and 301 downregulated in B16-CCRL2^OE^ spheroids, and 1151 genes upregulated and 611 downregulated in B16-CCRL2^KO^ spheroids compared to controls (fold change > 2; adjusted *p*-value < 0.05) (Figure 6B,C). Notably, there was limited overlap between the gene sets altered in the two conditions, suggesting non-redundant, context-dependent effects of CCRL2 modulation (Figure 6D). To assess the biological relevance of these transcriptional changes, Gene Ontology (GO) enrichment analysis was performed using g:Profiler. Genes upregulated In CCRL2^OE^ spheroids were enriched for processes such as extracellular matrix organization, cell adhesion, cell junction assembly, and protein binding (Figure 6E; Appendix A), while downregulated genes were associated with system development, catalytic activity, and protein binding (Appendix A; Appendix A). In contrast, CCRL2^KO^ spheroids showed downregulation of genes involved in cytoskeletal organization, homophilic cell adhesion, multicellular development, cytoplasmic structure, and the cell cycle (Figure 6E; Appendix A), while upregulated genes were enriched in immune-related functions, membrane localization, ion binding, and response to stimuli (Appendix A; Appendix A). These transcriptomic patterns mirror the observed structural differences between spheroids and suggest a regulatory role for CCRL2 in controlling tumor architecture via transcriptional modulation of adhesion and signaling pathways. To identify potential upstream regulators, we performed Ingenuity Pathway Analysis (IPA), focusing on membrane-associated signaling components. Given that CCRL2 is a non-signaling atypical chemokine receptor, we hypothesized that its impact may occur through functional interactions with other transmembrane receptors. IPA identified several deregulated surface receptors, including IFNAR1, IFNAR2, TLR3, TLR4, PTGER4, and FZD9 (Figure 6F; Appendix A), many of which are central to immune and inflammatory signaling. Supporting these observations, previous studies have shown that CCRL2 can modulate innate immune signaling via direct interactions with receptors such as TLR4, enhancing its plasma membrane stability and promoting MyD88-dependent NF-κB signaling in tumor-associated macrophages [27] (Figure 6G). Furthermore, CCRL2 has been implicated in activating cell-intrinsic IFN-γ/STAT1 signaling in TP53-mutant AML, via JAK2-dependent and independent pathways, even in the absence of exogenous IFN-γ [28] (Figure 6G). Consistent with this, our transcriptomic data revealed that both CCRL2^OE^ and CCRL2^KO^ spheroids exhibited activation of TLR4 and IFNAR1 signaling, although with divergent downstream outputs. TLR4 and IFNAR1 activation was predicted in both CCRL2^OE^ (Z = 2.635 and Z = 2.761, respectively) and CCRL2^KO^ (Z = 3.366 and Z = 4.450, respectively) spheroids (Figure 6F). However, the sets of downstream transcriptional targets differed substantially, indicating a context-dependent rewiring of TLR4- and IFNAR-mediated effector programs depending on CCRL2 expression status. To further explore this, we used IPA-based analysis to identify transcription factors and signaling intermediates (Figure 6H). Both conditions demonstrated activation of key immune regulators, including STAT1, IRF3, NF-κB, and AP-1, although with differing magnitudes. CCRL2^KO^ spheroids exhibited stronger activation of IFNG, STAT1, IRF3, and MYD88, suggesting a more pronounced interferon and TLR-driven immune response. In contrast, CCRL2^OE^ spheroids showed greater activation of NF-κB, AP-1, ERK, JNK, and p38 MAPK (Figure 6H; Appendix A), indicating preferential engagement of MAPK and pro-inflammatory pathways. Heatmap analysis confirmed these trends, revealing an inverse gene expression pattern between CCRL2^OE^ and CCRL2^KO^ samples relative to B16 controls (Figure 6I). Genes upregulated in CCRL2^OE^ spheroids were consistently downregulated in CCRL2^KO^ spheroids, and vice versa. This bidirectional regulatory pattern was particularly evident in STAT1 and IRF3 downstream signatures, further supporting the hypothesis that CCRL2 serves as a key modulator of innate immune gene programs in melanoma cells. Together, these findings demonstrate that CCRL2 governs transcriptional programs in melanoma cells through bidirectional modulation of TLR4 and IFNAR signaling pathways. Acting through receptor crosstalk rather than intrinsic signaling, CCRL2 orchestrates the balance between interferon-mediated and MAPK-driven immune responses, ultimately shaping both the structural and immunological properties of melanoma spheroids.

## 4. Discussion

In this study, we investigated the cell-intrinsic role of CCRL2 in tumor cell behavior, independent of its established function in immune surveillance and chemerin/CMKLR1 signaling. Building on our previous work demonstrating that CCRL2 modulates tumor growth in vivo through the chemerin/CMKLR1 axis [29], we now show that CCRL2 also influences tumor architecture and inflammatory signaling from within the tumor cells themselves. Previous studies have explored the impact of CCRL2 expression on various tumor cell lines. For example, Yin et al. reported that CCRL2 overexpression did not alter proliferation in glioblastoma cells but enhanced their migration and invasion [21]. Similarly, Akram et al. found that CCRL2 knockdown had mild effects on the proliferation and survival of prostate cancer cells [22], while Wang et al. showed that CCRL2 overexpression suppressed breast cancer cell growth both in vitro and in vivo [34]. These findings underscore the cell-type specific effects of CCRL2 and prompted us to examine its functional impact in murine melanoma and lung carcinoma models. In our system, CCRL2 overexpression or knockout did not significantly affect the proliferation, migration, or clonogenic capacity of B16 or LLC cells in 2D culture, but it did influence the size and morphology of the colonies they formed. Moreover, using B16 spheroids as a model of 3D tumor growth, we uncovered a clear role for CCRL2 in modulating spheroid architecture. CCRL2-overexpressing cells formed compact, cohesive spheroids, while CCRL2-deficient cells formed loosely aggregated, disorganized structures. These phenotypic differences were consistent across multiple culture conditions, suggesting that CCRL2 intrinsically governs tumor cell organization. Transcriptomic analysis provided insight into the molecular mechanisms underlying the distinct spheroid phenotypes. CCRL2 was found to modulate innate immune signaling networks, with a particular impact on pathways downstream of TLR4 and IFN-γ/STAT1. Interestingly, both CCRL2 overexpression and knockout resulted in predicted activation of TLR4 and IFNAR1 signaling; however, the downstream transcriptional programs differed markedly between the two conditions. This suggests a context-dependent rewiring of effector responses, whereby CCRL2 expression shapes the qualitative output of shared immune signaling axes.

These observations are particularly intriguing given that CCRL2 is an atypical chemokine receptor lacking canonical signaling motifs. Rather than initiating signaling directly, CCRL2 may function indirectly, by stabilizing receptor complexes, modulating membrane microdomains, or influencing receptor trafficking. This is consistent with prior studies showing CCRL2’s interaction with TLR4 in macrophages, enhancing its stability and supporting NF-κB activation, and its involvement in IFN-γ/STAT1 signaling in hematologic malignancies, even in the absence of exogenous cytokine. Our functional data suggest that CCRL2 acts not as a binary regulator but rather as a tunable scaffold, which biases the output of shared signaling pathways toward distinct outcomes. In CCRL2^OE^ cells, this favors programs related to cell adhesion, homeostasis, and structural integrity, whereas in CCRL2^KO^ cells, it shifts toward pro-inflammatory gene expression and potential motility. This bidirectional regulatory role reflects the complexity of CCRL2’s function and positions it alongside other atypical chemokine receptors as a modulator of receptor crosstalk and signaling balance.

## 5. Conclusions

In summary, our study uncovers a novel, cell-autonomous function of CCRL2 in controlling tumor spheroid morphology and inflammatory gene expression. By differentially modulating the TLR4 and IFN-γ/STAT1 signaling axes, CCRL2 acts as a context-specific regulator of innate immune responses and tumor architecture. These findings extend CCRL2’s relevance beyond immune surveillance, highlighting its potential as a multifunctional modulator and therapeutic target in both immune-competent and immune-deprived tumor settings. Further studies across diverse tumor cell lines and cancer types will be essential to determine whether this CCRL2-driven regulatory mechanism is broadly conserved or contextually restricted.

## Figures and Tables

**Figure 1 biomolecules-15-01150-f001:**
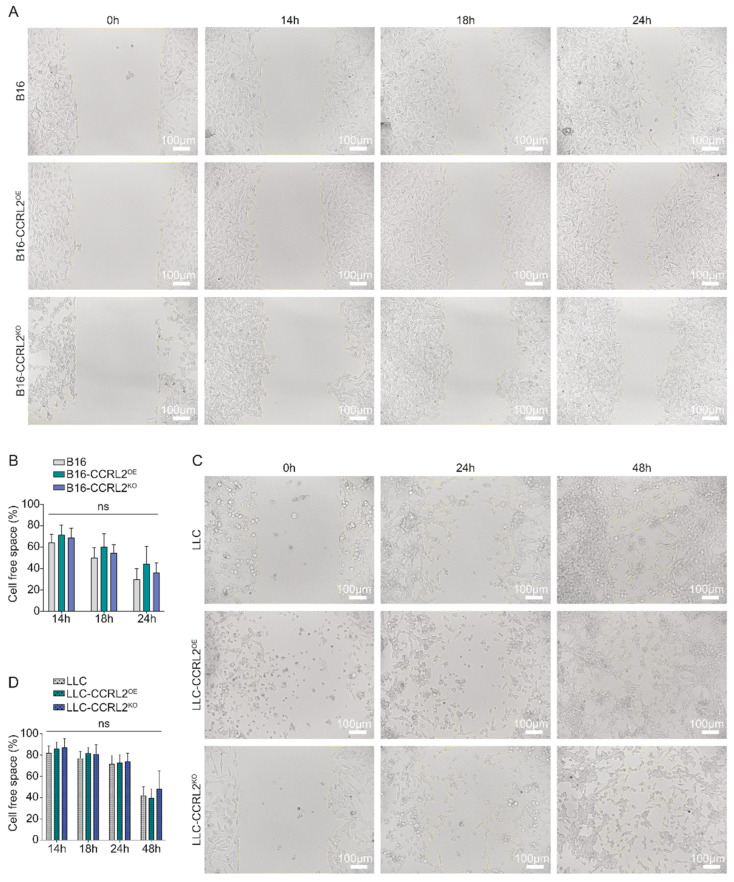
Effect of CCRL2 expression on the migratory capacity of B16 and LLC tumor cell lines. (**A**,**C**) Representative images of wound healing assays performed on B16 melanoma cells (**A**) and LLC cells (**C**) either overexpressing CCRL2 (B16/LLC-CCRL2^OE^) or knocked out for Ccrl2 (B16/LLC-CCRL2^KO^), compared to the controls. Standardized wound areas were generated using culture inserts, and migration was monitored over time. Images include overlays highlighting the remaining cell-free space. (**B**,**D**) Quantification of the remaining cell-free space (%) at each time point, calculated as (Cell-free area/Cell-free area at T = 0 h) × 100%. Data are presented as mean ± SEM from three independent experiments (ns: not significant). Migration was analyzed using ImageJ software. Scale bars: 100 μm.

**Figure 2 biomolecules-15-01150-f002:**
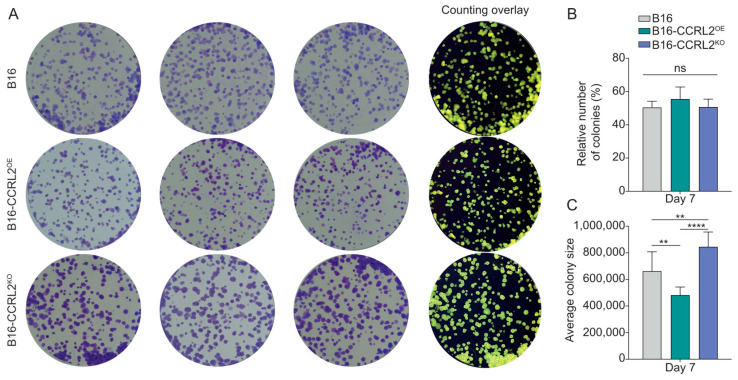
Impact of CCRL2 expression on clonogenic potential and colony architecture in B16 cells. (**A**) Representative images of crystal violet-stained colonies formed by control B16 cells, CCRL2-overexpressing (B16-CCRL2^OE^), and CCRL2-knockout (B16-CCRL2^KO^) cells after plating 500 cells and culturing for one week. The rightmost column shows image analysis overlays generated using ImageJ software for colony counting. (**B**) Quantification of colony formation efficiency calculated as (Number of colonies formed/Number of cells seeded) × 100%. Data are presented as mean ± SD from six independent experiments. No statistically significant differences were observed among the groups (ns: not significant). (**C**) Quantification of average colony size (µm^2^) on day 7. Data represent mean ± SD from four independent experiments. One-way ANOVA with Tukey’s post hoc test (** *p* < 0.01, **** *p* < 0.0001).

**Figure 3 biomolecules-15-01150-f003:**
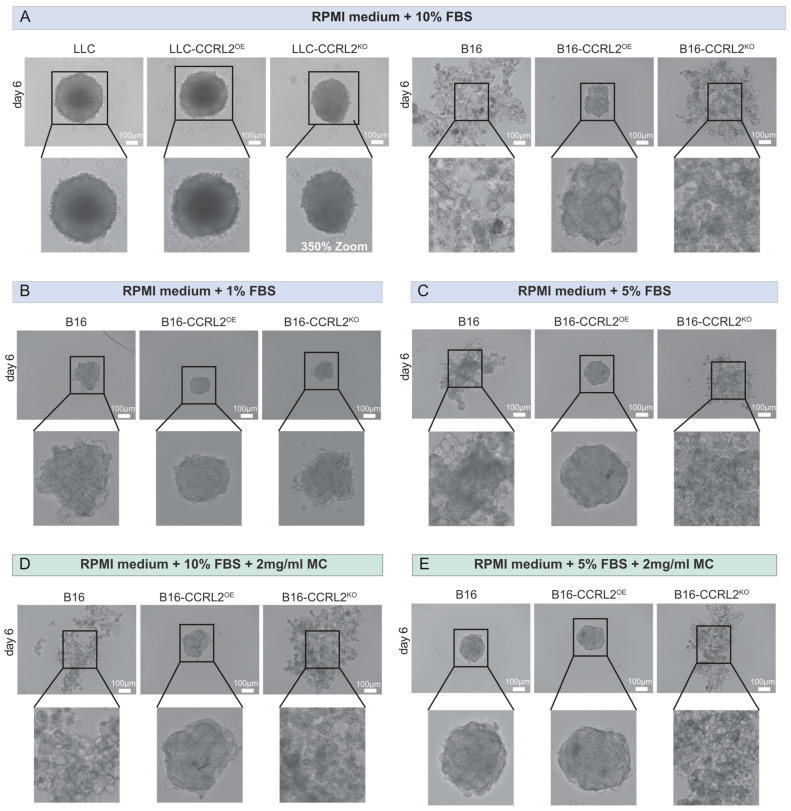
Effect of CCRL2 expression on 3D spheroid formation and morphology in LLC and B16 tumor cell lines under different culture conditions. (**A**) Representative images of spheroids formed by LLC and B16 cells at day 6 (control, CCRL2-overexpressing, and CCRL2-knockout) cultured in RPMI medium supplemented with 10% FBS. (**B**,**C**) B16 spheroids grown in RPMI medium containing 1% (B) or 5% (**C**) FBS. (**D**,**E**) B16 spheroids cultured in RPMI medium with either 10% (**D**) or 5% (**E**) FBS supplemented with 2 mg/mL methylcellulose (MC), used to enhance aggregation. Magnified insets highlight spheroid structure and compactness. Scale bars: 100 μm. Images are representative of at least three independent experiments.

**Figure 4 biomolecules-15-01150-f004:**
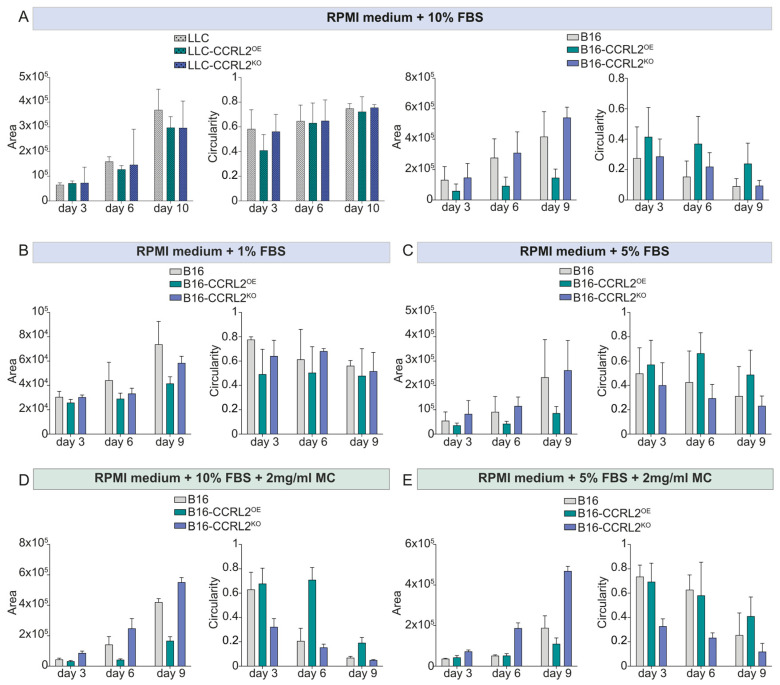
Quantitative analysis of spheroid growth and morphology in LLC and B16 cells under different culture conditions. Spheroid area (µm^2^) and circularity (a measure of compactness, with values closer to 1.0 indicating a more spherical morphology) were quantified using ImageJ in parental (control), CCRL2-overexpressing (CCRL2^OE^), and CCRL2-knockout (CCRL2^KO^) variants of LLC and B16 melanoma cell lines. Cells were cultured under varying serum concentrations (panels **A**–**C**) or with methylcellulose supplementation (panels **D**,**E**). Quantifications were performed at days 3, 6, 9, or 10, depending on the condition. All corresponding statistical analyses are provided in Appendix A.

**Figure 5 biomolecules-15-01150-f005:**
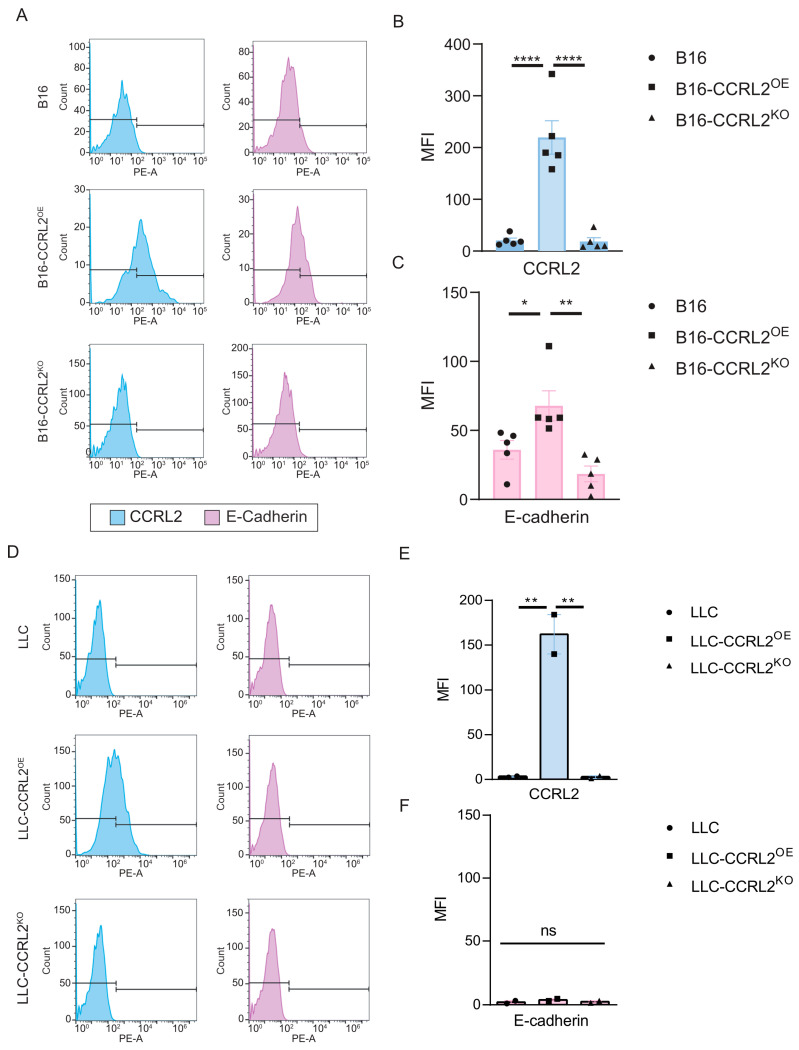
CCRL2 and E-cadherin expression in tumor spheroids derived from B16 and LLC cells. Flow cytometry analysis of CCRL2 (blue histograms) and E-cadherin (purple histograms) expression in (**A**) B16, B16-CCRL2^OE^, and B16-CCRL2^KO^ spheroids and in (**D**) LLC, LLC-CCRL2^OE^, and LLC-CCRL2^KO^ spheroids. Tumor spheroids were cultured in medium containing 10% FBS and harvested on day 6 for analysis. (**B**,**E**) Quantification of CCRL2 expression levels based on mean fluorescence intensity (MFI) in B16 and LLC spheroids, respectively. (**C**,**F**) Quantification of E-cadherin MFI in the same conditions. Data shown are representative of four independent experiments for B16 and two for LLC, with consistent results. Statistical significance was determined using one-way ANOVA followed by Tukey’s post hoc test (* *p* < 0.05, ** *p* < 0.01, **** *p* < 0.0001; ns: not significant).

**Figure 6 biomolecules-15-01150-f006:**
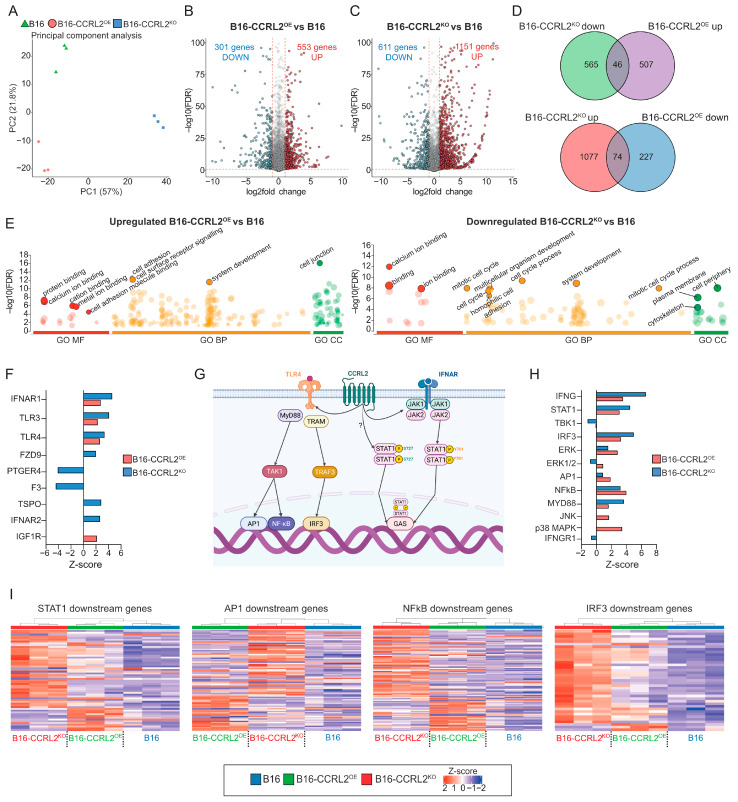
CCRL2 expression modulates distinct transcriptional programs in B16 spheroids. (**A**) Principal component analysis (PCA) of bulk RNA sequencing data from B16 (green), B16-CCRL2^OE^ (red), and B16-CCRL2^KO^ (blue) spheroids cultured in 10% FBS and collected at day 6. Each point represents an individual biological replicate. (**B**,**C**) Volcano plots displaying differentially expressed genes between B16-CCRL2^OE^ vs. B16 (**B**) and B16-CCRL2^KO^ vs. B16 (**C**). Red dots represent significantly upregulated genes, and blue dots represent significantly downregulated genes (adjusted *p* < 0.05, |log_2_FC| > 1). Vertical and horizontal dashed lines mark significance and fold change thresholds. (**D**) Venn diagrams showing the overlap between DEGs in the two comparisons for both upregulated and downregulated genes. (**E**) Gene Ontology (GO) enrichment analysis of DEGs performed using g:Profiler. Each dot represents an enriched GO term grouped by domain: Molecular Function (GO:MF, red), Biological Process (GO:BP, orange), and Cellular Component (GO:CC, green). The x-axis indicates GO categories, and the y-axis shows –log_10_ of the adjusted *p*-value. The size of each dot reflects the size of the gene set associated with that term. (**F**) Bar graph showing the expression Z-scores of selected transmembrane receptors and G protein-coupled receptors (GPCRs) identified through Ingenuity Pathway Analysis (IPA). Receptors were selected based on both predicted upstream regulatory activity and the availability of a corresponding gene expression log ratio from RNA-seq data. (**G**) Schematic representation of CCRL2-related signaling interactions based on the published literature, illustrating crosstalk with TLR4, and IFNAR pathways. Downstream transcription factors such as STAT1, AP-1, NF-κB, and IRF3 are shown. The arrows indicate the stepwise progression of the signaling pathway. The question mark represents an unknown or independent pathway. Created with BioRender.com https://app.biorender.com/illustrations/67fcf56dd06583caecb642c5?slideId=79a3abef-dc31-4011-a16b-42bb183051d4 (accessed on 25 July 2025) (**H**) Z-score plot of downstream effectors and transcription factors identified through IPA downstream analysis as deregulated in B16-CCRL2^OE^ and B16-CCRL2^KO^ cells. Only molecules associated with TLR4 and IFNAR signaling pathways were included. (**I**) Heatmaps showing expression levels of downstream target genes regulated by STAT1, AP-1, NF-κB, and IRF3 transcription factors in B16-CCRL2^OE^, and B16-CCRL2^KO^ cells compared to control. Each row represents a gene and each column a biological replicate.

## Data Availability

The data of RNA sequencing have been deposited in NCBI’s Gene Expression Omnibus and are accessible through GEO Series accession number GSE299192. The source data for all figures is available as a Appendix A. The data supporting all figures in this paper can be found at the Open Science Framework repository https://osf.io/tq29g/ (accessed on 29 May 2025).

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
