# Peer review of "Atypical Chemokine Receptor CCRL2 Shapes Tumor Spheroid Structure and Immune Signaling in Melanoma"

_biomolecules, 2025, doi:10.3390/biom15081150_

Round 1

Reviewer 1 Report

Comments and Suggestions for Authors

The manuscript investigates the potential tumor-intrinsic functions of the atypical receptor CCRL2 in B16F0 melanoma and LLC carcinoma cells. No differences were observed in proliferation, migration, or clonogenicity between CCRL2-overexpressing and CCRL2 knockout (KO) B16 and LLC cell lines. However, the authors demonstrate that CCRL2 modulates the spheroid-forming capacity of B16 melanoma cells, but not LLC carcinoma cells, indicating a cell-type-dependent effect. Furthermore, CCRL2 expression correlates with E-Cadherin levels only in B16 spheroids, not in LLC, and these findings are supported by CCRL2-dependent transcriptional programs in B16 tumors.

The manuscript is potentially of interest; however, in several cases, the conclusions are not fully supported by the transcriptional or morphological data provided. Additional functional validation of key transcriptomic hits is necessary to strengthen the study.

Specific comments:

  • Figures 1A and 1C: These figures are not very clear and, given that they show negative results, could be moved to the Supplementary section. Including time point 0 in the graphs would improve interpretation.
  • Figures 2A and S1A: The larger colonies observed in B16-CCRL2KO cells should be quantified.
  • Figure 3A: The descriptions of the left and right panels (lines 290 and 294) appear to be mixed up and should be corrected.
  • Figure 3: Spheroid structures should be quantitatively evaluated by measuring diameter, area, and perimeter. The extensive discussion of experimental conditions (serum concentration, methylcellulose supplementation) is somewhat confusing and could be moved to the Supplementary section.
  • It is intriguing that CCRL2 affects cell organization only in B16 spheroids, not in LLC, and that CCRL2’s cell-autonomous effects are enhanced by overexpression. However, since basal CCRL2 expression in B16 cells appears sufficient to maintain spheroid structure at day 6 (Figure 3)—even though CCRL2 and E-Cadherin surface levels are reportedly low (Figure 4)—the authors are encouraged to provide additional data on both surface and intracellular levels of CCRL2 and E-Cadherin across different spheroid culture time points. Histograms showing MFI (mean fluorescence intensity) values and appropriate statistical analyses would significantly help readers interpret CCRL2’s role in the B16 model.
  • Cell viability at later stages: Since B16 spheroids dissociate at days 6 and 9, it is important to determine whether the cells remain viable. Please provide data supporting their viability.
  • Line 304: The reference should be to Figure 3B, not 3C.
  • RNA-seq timing: The RNA sequencing experiment is a key part of the manuscript. However, the time point selected (day 6) is problematic, as CCRL2KO B16 spheroids are already disaggregated at this stage, which likely affects the transcriptomic profile. A more appropriate time point would be day 3, which might capture early transcriptional changes preceding the dramatic phenotype observed at day 6. Additionally, a control experiment inducing spheroid disaggregation independently of CCRL2 (e.g., low-dose protease treatment) could help distinguish CCRL2-specific effects from general disaggregation-induced changes.
  • Validation of transcriptomic data: Candidate genes from RNA-seq should be validated by qRT-PCR. Furthermore, surface expression of IFNARs and TLR4 should be assessed, and activation of the STAT1 and NF-κB pathways confirmed via Western blot or flow cytometry.

Author Response

Summary:

We sincerely thank the reviewers for their constructive and thoughtful comments, which have helped us to further refine and improve our manuscript. We fully acknowledge the value of validating the transcriptomic findings at the protein level, including the assessment of IFNAR and TLR4 surface expression, activation of downstream pathways such as STAT1 and NF-κB, and confirmation of key transcription factors and candidate genes through qRT-PCR, Western blotting, or flow cytometry. These additional experiments would undoubtedly strengthen the mechanistic conclusions of our study. However, this work was conducted as part of my doctoral research, which I completed in 2021, and it represents the final part of a broader investigation into the role of CCRL2 in tumor biology. The present study builds directly upon our previously published work, “Expression of CCRL2 Inhibits Tumor Growth by Concentrating Chemerin and Inhibiting Neoangiogenesis” (DOI: 10.3390/cancers13195000), in which we demonstrated the anti-tumor activity of CCRL2 in vivo through modulation of the chemerin/CMKLR1 axis. Importantly, our earlier findings suggested that CCRL2’s effects were not fully dependent on this pathway, prompting us to explore potential cell-intrinsic roles for CCRL2 in tumor cells. In this follow-up study, we therefore performed a focused, mechanistic analysis in a well-characterized melanoma model, integrating transcriptomic profiling with functional assays. Given the scope and constraints of the completed PhD project, we are unfortunately unable to conduct further validation experiments. Nonetheless, we are confident that the data presented offer valuable insights into CCRL2-mediated regulation of tumor cell behavior and lay a solid foundation for future investigation.

Reviewer 1

Figures 1A and 1C: These figures are not very clear and, given that they show negative results, could be moved to the Supplementary section. Including time point 0 in the graphs would improve interpretation.

We thank the reviewer for the suggestion. However, we believe that Figures 1A and 1C are best retained in the main manuscript, as they provide essential context for the study. While CCRL2 has previously been shown to modulate migration in certain tumor cell lines, our data demonstrate that this is not the case in the melanoma and carcinoma models used here. This negative finding is informative and helps delineate the context-specific effects of CCRL2, supporting the rationale for exploring alternative functional roles in subsequent sections of the manuscript.

Regarding time point 0, migration was quantified as the percentage of remaining cell-free area relative to T=0h, which is standardized to 100% [(cell-free area / area at T=0h) × 100%]. As such, the baseline (T=0h) is inherently represented in the data as the reference point for all subsequent measurements, as noted in the figure legend for clarity.

Figures 2A and S1A: The larger colonies observed in B16-CCRL2KO cells should be quantified.

We have now quantified the colony sizes in B16, B16-CCRL2OE and B16-CCRL2KO cells, and the corresponding data have been included in the revised Figures 2A and S1A, along with the appropriate quantification panels and updated figure legends.

Figure 3A: The descriptions of the left and right panels (lines 290 and 294) appear to be mixed up and should be corrected.

We have now corrected the text accordingly to ensure accurate referencing of each panel.

Figure 3: Spheroid structures should be quantitatively evaluated by measuring diameter, area, and perimeter.

While the morphological differences between spheroids were visually striking and reproducible, we fully agree that quantitative analysis provides greater robustness of the findings. In response, we have now performed a detailed quantification of spheroid area, circularity, diameter, and perimeter. The area and circularity data are now included in the revised main Figure 4, and the diameter and perimeter measurements have been added to the Supplementary Figures, with corresponding legends and explanations.

Figure 3: The extensive discussion of experimental conditions (serum concentration, methylcellulose supplementation) is somewhat confusing and could be moved to the Supplementary section.

We thank the reviewer for this suggestion. While we initially moved the corresponding images to the Supplementary section for clarity, we would like to note that Reviewer 2 did not raise any concerns regarding the inclusion of these experimental conditions in the main text. Given their relevance in supporting the interpretation of the CCRL2-dependent phenotype, we have decided to retain the discussion of serum concentration and methylcellulose supplementation in the main manuscript to provide appropriate context. To maintain clarity and focus, we now include only the day 6 representative images  (Figure 3) and quantification plots (Figure 4) (area and circularity) in the main figures, while detailed morphological images from additional time points have been moved to the Supplementary section.

Moreover, these experimental conditions also address the second reviewer’s suggestion regarding the use of protease treatment to test spheroid integrity. Rather than disrupting the spheroids, we applied well-established culture modifications (serum concentration and methylcellulose) to enhance spheroid compactness. The results clearly demonstrate that the observed phenotype is robust and CCRL2-dependent under varying structural conditions, thereby reinforcing the specificity and strength of the phenotype.

Figures 3-4: It is intriguing that CCRL2 affects cell organization only in B16 spheroids, not in LLC, and that CCRL2’s cell-autonomous effects are enhanced by overexpression. However, since basal CCRL2 expression in B16 cells appears sufficient to maintain spheroid structure at day 6 (Figure 3)—even though CCRL2 and E-Cadherin surface levels are reportedly low (Figure 4), the authors are encouraged to provide additional data on both surface and intracellular levels of CCRL2 and E-Cadherin across different spheroid culture time points. Histograms showing MFI (mean fluorescence intensity) values and appropriate statistical analyses would significantly help readers interpret CCRL2’s role in the B16 model.

We would like to clarify that CCRL2 expression in B16 control cells is undetectable under standard in vitro conditions. Despite extensive efforts, including culturing the cells in various media formulations in 2D, we were unable to detect CCRL2 expression at either the RNA or protein level, as assessed by qPCR and flow cytometry. This absence of detectable expression was a key rationale for generating and utilizing CCRL2-overexpressing B16 cell lines in this study. Importantly, we have shown in our previous work (DOI: 10.3390/cancers13195000), that CCRL2 expression is significantly upregulated in vivo following tumor implantation, likely in response to the inflammatory tumor microenvironment. In that context, CCRL2 expression was associated with reduced tumor growth, particularly in CCRL2-deficient hosts, due to the accumulation of chemerin within the tumor. These findings further support the relevance of studying CCRL2’s cell-intrinsic functions in a controlled model system.

In spheroid cultures, flow cytometry confirmed that B16 control and B16-CCRL2KO spheroids do not express detectable levels of CCRL2, whereas spheroids derived from CCRL2-overexpressing cells exhibit robust CCRL2 expression, was accompanied by higher levels of E-cadherin compared to the other groups, likely contributing to the phenotypic differences observed. We focused our analyses on day 6, when structural changes were most pronounced, and performed FACS experiment at day 11, which confirmed similar expression patterns. In response to the reviewer’s suggestion, we have now included MFI histograms in the main figure, along with appropriate statistical analysis, to better illustrate CCRL2 expression and support data interpretation.

Cell viability at later stages: Since B16 spheroids dissociate at days 6 and 9, it is important to determine whether the cells remain viable. Please provide data supporting their viability.

We thank the reviewer for raising this important point. However, we would like to clarify that B16 spheroids do not dissociate at days 6 or 9. The observed differences reflect variations in spheroid compactness and organization, rather than disintegration or cell death. In fact, the spheroids remain structurally intact and continue to grow beyond day 9, indicating sustained cell viability. To address the reviewer’s concern more directly, we have included flow cytometry scatter plots (Figure attached) from dissociated spheroids at day 6 (B16 control, CCRL2-OE, and CCRL2-KO). These data confirm the presence of a viable cell population across all conditions. As the CCRL2-overexpressing spheroids are smaller in size, they naturally yield fewer total cells. For flow cytometric analysis, several spheroids were pooled per condition to ensure sufficient cell numbers for staining.

Line 304: The reference should be to Figure 3B, not 3C.

We thank the reviewer for pointing this out. The reference has been corrected to Figure 3B in the revised manuscript.

Figure 5: RNA-seq timing: The RNA sequencing experiment is a key part of the manuscript. However, the time point selected (day 6) is problematic, as CCRL2KO B16 spheroids are already disaggregated at this stage, which likely affects the transcriptomic profile. A more appropriate time point would be day 3, which might capture early transcriptional changes preceding the dramatic phenotype observed at day 6. Additionally, a control experiment inducing spheroid disaggregation independently of CCRL2 (e.g., low-dose protease treatment) could help distinguish CCRL2-specific effects from general disaggregation-induced changes.

We thank the reviewer for this comment. B16-CCRL2KO spheroids do not disaggregate at day 6. Rather, they exhibit a looser, less compact morphology from as early as day 1, commonly referred to as a “grape-like” phenotype. This structural pattern is well described in 3D tumor models and is typically associated with reduced cell–cell adhesion and a loss of epithelial characteristics.

As described by Kenny et al. (Molecular Oncology, 2007; https://doi.org/10.1016/j.molonc.2007.02.004), grape-like spheroids are characterized by reduced cell–cell adhesion and lower E-cadherin expression compared to other morphological classes. These structures typically form loosely associated colonies, often derived from metastatic cell lines, and reflect a diminished epithelial phenotype. Interestingly, while proliferation (as assessed by Ki-67 staining) did not correlate with morphology, E-cadherin levels were consistently lower in grape-like colonies, likely due to their limited intercellular cohesion. Their study further demonstrated that spheroid morphology correlates with distinct gene and protein expression profiles, despite no direct association with proliferation levels, underscoring the biological relevance of spheroid architecture.

 (https://doi.org/10.1016/j.molonc.2007.02.004)

Similarly, Gorzalczany et al. (2023) (Anticancer Research, https://doi.org/10.21873/anticanres.16290) used B16-F10 melanoma spheroids as a 3D model to investigate the effects of mast cells and mast cell-conditioned medium. These B16 spheroids formed loosely aggregated, grape-like structures, grew robustly under standard culture conditions without any treatment, and maintained the expression of melanoma-associated genes. A significant reduction in spheroid growth was only observed upon exposure to mast cells or mast cell-conditioned medium.

In our study, we selected day 6 for RNA-seq because it provided the most reliable and reproducible window to capture CCRL2-dependent transcriptional changes. Although morphological differences between conditions were already noticeable by day 3, spheroid structures at that earlier time point showed more variability across replicates, particularly in the control group. By day 6, the spheroids exhibited more consistent and stable morphologies while maintaining viability, ensuring that observed phenotypic divergence was robust and representative. This allowed us to maximize the interpretability and biological relevance of the transcriptomic data.

We appreciate the suggestion to include a protease-induced disaggregation control; however, our data, supported by previous studies on B16 melanoma and other cancer cell lines, indicate that the observed “grape-like” morphology represents a stable, loosely aggregated spheroid phenotype rather than true disaggregation. This architecture has been consistently described as a characteristic growth pattern in melanoma spheroids under certain conditions, where cells remain viable and maintain melanoma-specific gene expression despite reduced compactness. Therefore, we believe the changes we observed are unlikely to reflect nonspecific disaggregation responses, and a protease-induced dissociation control would not directly address the specific features of this grape-like spheroid architecture.

Validation of transcriptomic data: Candidate genes from RNA-seq should be validated by qRT-PCR. Furthermore, surface expression of IFNARs and TLR4 should be assessed, and activation of the STAT1 and NF-κB pathways confirmed via Western blot or flow cytometry.

We fully agree that validating key transcriptomic findings at the protein level would further strengthen the study. However, as this study was conducted within the scope of a completed PhD project, we are unable to carry out additional experiments at this stage. Notably, our previous work showed that CCRL2’s activity is not fully dependent on the Chemerin/CMKLR1 pathway, motivating this follow-up study to explore its potential cell-intrinsic roles. To that end, we performed a focused mechanistic analysis integrating transcriptomic and functional data in a well-characterized melanoma model. While we cannot extend the study further, we believe the results offer robust insights into CCRL2’s autonomous function and form a strong basis for future investigation.

Reviewer 2 Report

Comments and Suggestions for Authors

In this study, Diana Al Delbany et al. demonstrated that the atypical chemokine receptor CCRL2 influences tumor spheroid structure. The authors also discussed the effect of CCRL2 in immune signaling in melanoma through RNA sequencing and transcriptome analysis in the B16 cell line, overexpression or knockout of CCRL2.

The overall findings of this study are logically presented. The spheroid formation approach is very interesting in my view. However, it remains unclear about the detailed mechanisms, especially for confirming the downstream signaling implied by the RNA-Seq. Furthermore, essential quantification and control measures were absent. Please check the listed comments below:

Major Points

1. For figures 2-4, it would be helpful if the author could also provide quantification data to assist the reader in directly drawing conclusions.

2. It is necessary to demonstrate the efficacy of CCRL2 overexpression and knockout, particularly for the RNA-Seq sample.

3. The pathway analysis in Figure 5F is intriguing. However, as one of the most significant conclusions of this study, it is crucial to confirm the receptors at the protein level and whether they can localize on the cell surface to enhance the stimulus response.

4. Similarly, it is necessary to demonstrate the TFs at the protein level in Figure 5H. Furthermore, I am curious about how to explain the heatmap results in Figure 5I, as the cells did not receive any specific stimulation. Even assuming that TFs are indeed regulated at the protein level, they still require the necessary signals to regulate nuclear entry or downstream signaling.

Minor Points

1. For figure 2, using techniques such as CFES or BrdU labeling could be helpful in making the conclusion that CCRL2 does not affect the target cell line proliferation.

2. For the heatmap in Figure 5I, I am wondering about the distinction between the upregulated and downregulated subgroups between CCRL2 overexpression and knockout.

3. It would be helpful to label the groups (B16, OE, KO) in the figure more clearly, as the color bar at the top of the heatmap is not obvious. Additionally, it is better to keep the order consistency (AP1 vs others).

Author Response

Summary:

We sincerely thank the reviewers for their constructive and thoughtful comments, which have helped us to further refine and improve our manuscript. We fully acknowledge the value of validating the transcriptomic findings at the protein level, including the assessment of IFNAR and TLR4 surface expression, activation of downstream pathways such as STAT1 and NF-κB, and confirmation of key transcription factors and candidate genes through qRT-PCR, Western blotting, or flow cytometry. These additional experiments would undoubtedly strengthen the mechanistic conclusions of our study. However, this work was conducted as part of my doctoral research, which I completed in 2021, and it represents the final part of a broader investigation into the role of CCRL2 in tumor biology. The present study builds directly upon our previously published work, “Expression of CCRL2 Inhibits Tumor Growth by Concentrating Chemerin and Inhibiting Neoangiogenesis” (DOI: 10.3390/cancers13195000), in which we demonstrated the anti-tumor activity of CCRL2 in vivo through modulation of the chemerin/CMKLR1 axis. Importantly, our earlier findings suggested that CCRL2’s effects were not fully dependent on this pathway, prompting us to explore potential cell-intrinsic roles for CCRL2 in tumor cells. In this follow-up study, we therefore performed a focused, mechanistic analysis in a well-characterized melanoma model, integrating transcriptomic profiling with functional assays. Given the scope and constraints of the completed PhD project, we are unfortunately unable to conduct further validation experiments. Nonetheless, we are confident that the data presented offer valuable insights into CCRL2-mediated regulation of tumor cell behavior and lay a solid foundation for future investigation.

Reviewer 2

  1. For figures 2-4, it would be helpful if the author could also provide quantification data to assist the reader in directly drawing conclusions.

We thank the reviewer for pointing this out. We have now quantified the colony sizes in B16, B16-CCRL2OE and B16-CCRL2KO cells, and the corresponding data have been included in the revised Figures 2A and S1A, along with the appropriate quantification panels and updated figure legends.

While the morphological differences between spheroids were visually striking and reproducible, we fully agree that quantitative analysis provides greater robustness of the findings. In response, we have now performed a detailed quantification of spheroid area, circularity, diameter, and perimeter. The area and circularity data are now included in the revised main Figure 3, and the diameter and perimeter measurements have been added to the Supplementary Figures, with corresponding legends and explanations.

We have now included MFI histograms in the figure 4, along with appropriate statistical analysis, to better illustrate CCRL2 expression and support data interpretation.

  1. It is necessary to demonstrate the efficacy of CCRL2 overexpression and knockout, particularly for the RNA-Seq sample.

We appreciate the reviewer’s concern regarding the validation of CCRL2 modulation. The efficacy of CCRL2 knockout and overexpression was thoroughly characterized in our previous work (DOI: 10.3390/cancers13195000), which included:

CRISPR/Cas9-based knockout validation: All edited clones for both LLC and B16 were initially screened using a heteroduplex mobility assay (Figure 1A-B), from which five clones per cell line were selected for further sequencing. For these selected B16 and LLC clones, PCR amplicons were cloned into a plasmid vector, and individual inserts were sequenced. Sequencing revealed that the sgRNA pair targeting the 5′ coding region of CCRL2 induced various mutations, including premature stop codons in both alleles (Figure 1A-B). Based on these results, B16 clones 12 and 16 and LLC clones 11 and 27 were confirmed as CCRL2-null lines and used for subsequent experiments.

Figure 1. Validation of CCRL2 knockout in LLC (A) and B16 (B) tumor cells. A heteroduplex mobility assay was performed by PCR amplification of the CCRL2 target region from genomic DNA of individual clones, with primers flanking the sgRNA sites. Amplicons were separated on a 15% polyacrylamide gel, where mutated clones showed multiple heteroduplex bands (red arrows), while control and non-mutated clones showed a single band (blue arrows); the expected wild-type amplicon size (239 bp) is indicated. Selected clones were then sequenced after cloning PCR products into the pCRII-TOPO vector, confirming mutations including deletions and insertions relative to the wild-type sequence. In these sequences, the start codon is bold and underlined, target genomic regions are shown in blue, PAM sites in green, and indels in red, with the size of each indicated to the right.

Flow cytometry analysis: We assessed surface CCRL2 expression by FACS to confirm protein-level knockout in the edited clones (Figure 2 attached).

PCR analysis: CCRL2 overexpression was verified by PCR amplification of the transgene as well as by flow cytometry for protein surface expression (Figure 2 attached).

Figure 2. Characterization of CCRL2 expression in B16 and LLC tumor cell lines. (A) and (B) FACS analysis of CCRL2 surface expression in B16 and LLC cells, respectively, including control cells, CCRL2-overexpressing clones, and CCRL2-knockout clones generated by CRISPR/Cas9. The grey profile shows staining with an isotype-matched control antibody. Data are representative of six independent experiments with similar results. Additionally, PCR analysis confirmed CCRL2 overexpression in these clones compared to control cells.

For the RNA-Seq samples specifically, these same validated CCRL2-overexpressing and knockout clones were used, ensuring consistency between functional experiments and transcriptomic profiling.

  1. The pathway analysis in Figure 5F is intriguing. However, as one of the most significant conclusions of this study, it is crucial to confirm the receptors at the protein level and whether they can localize on the cell surface to enhance the stimulus response.

We thank the reviewer for this important observation. We fully agree that confirming receptor expression and surface localization at the protein level would provide valuable mechanistic support. However, as this study was conducted within the scope of a completed PhD project, we are unable to perform additional experiments at this stage.

  1. Similarly, it is necessary to demonstrate the TFs at the protein level in Figure 5H. Furthermore, I am curious about how to explain the heatmap results in Figure 5I, as the cells did not receive any specific stimulation. Even assuming that TFs are indeed regulated at the protein level, they still require the necessary signals to regulate nuclear entry or downstream signaling.

We thank the reviewer for raising this important point. Indeed, transcription factors (TFs) such as STAT1, NF-κB, and AP-1 require upstream signaling events to become functionally active, including nuclear translocation and DNA binding. In our system, while no exogenous stimulation was applied, the B16 melanoma spheroid cultures provide an environment rich in endogenous activating signals. Specifically, B16 tumor cells are known to secrete a variety of cytokines and damage-associated molecular patterns (DAMPs), including HMGB1, S100 proteins, and heat shock proteins, that serve as ligands for pattern recognition receptors such as TLR4. These tumor-derived ligands have been shown to activate TLR4 signaling both in immune and tumor cells under unstimulated conditions (Erridge, J Leukoc Biol, 2010; Nagaraj et al., J Immunol, 2010).

Nonetheless, our findings are consistent with previously published data. In particular, our recent study (Zhou et al., Nat Commun, 2022) demonstrated that CCRL2 directly interacts with TLR4 and plays a critical role in retaining its surface expression, thereby amplifying Myd88–NF-κB signaling in macrophages. This effect was shown using B16-conditioned medium, indicating that tumor-derived endogenous ligands are sufficient to activate this pathway without external stimulation. This supports the plausibility of the observed transcriptional signatures in our 3D melanoma spheroids.

Furthermore, CCRL2 has been implicated in promoting cell-intrinsic IFN-γ/STAT1 signaling in TP53-mutant acute myeloid leukemia, even in the absence of exogenous IFN-γ, through both JAK2-dependent and -independent mechanisms (Naji et al., bioRxiv, 2025; doi:10.1101/2025.02.21.639304). These findings collectively support the idea that CCRL2 modulates receptor signaling and downstream TF activity independently of external stimulation.

While we fully agree that protein-level validation of receptor expression and TF activation would further strengthen the study, this work was conducted as part of a completed PhD project, and we are unfortunately unable to pursue additional experimental validation. Nonetheless, we believe the transcriptomic data, together with the functional phenotypes and literature evidence, provide a compelling rationale for CCRL2-dependent regulation of inflammatory signaling pathways.

 Minor Points

  1. For figure 2, using techniques such as CFES or BrdU labeling could be helpful in making the conclusion that CCRL2 does not affect the target cell line proliferation.

We thank the reviewer for this valuable suggestion. In our previous work, we extensively characterized the in vitro proliferation of these cell lines using well-established methods, including MTT assays, growth curve analyses, and doubling time measurements (Figure 3 attached), which consistently showed no significant differences attributable to CCRL2 expression (DOI: 10.3390/cancers13195000).

While CFSE or BrdU labeling provide a single-cell perspective on proliferation, they serve a similar purpose to the MTT and doubling-time analyses in addressing the question of whether CCRL2 affects cell proliferation. Given the robust and reproducible results from these prior assays, we believe they sufficiently support the conclusion that CCRL2 does not impact the intrinsic proliferation rate of these cells in vitro.

It is worth noting that the differences in tumor proliferation observed in vivo are more likely due to CCRL2’s role in modulating the tumor microenvironment, rather than intrinsic differences in tumor cell proliferation behavior under controlled in vitro conditions.

Figure 3. Overexpression or knockout of CCRL2 do not affect the proliferation rate of B16 and LLC clones. The proliferation of selected clones of B16 (A) and LLC cells (B) overexpressing CCRL2 or knockout for the gene was compared to that of WT cells by the MTT assay. The values were normalized to the average for WT cells at each time point. The data (mean ± SEM) represent the pool of three independent experiments with six wells per condition in each experiment. (C, D) The doubling time of the same clones was calculated by counting the cells at different time points and using the formula Td = (t2-t1) * log(2)/log(q2/q1), where q1 represents the number of cells at time t1 and q2 represents the number of cells at time t2. (E, F) The proliferation rate was also determined by counting the cells on days 1, 2, and 3 and normalizing the values according to the number of cells seeded on day 0. The data (mean ± SEM) are representative of 2 independent experiments with 3 wells per condition. (DOI: 10.3390/cancers13195000).

  1. For the heatmap in Figure 5I, I am wondering about the distinction between the upregulated and downregulated subgroups between CCRL2 overexpression and knockout.

The heatmaps in Figure 5I presents a representative subset of differentially expressed transcription factors, selected to highlight the most notable differences between CCRL2-overexpressing and knockout cells. Due to space and visualization constraints, it was not feasible to display all regulated genes within the heatmap. However, a complete list of upregulated and downregulated genes is provided in the accompanying Excel files (supplementary tables S4 and S5) for full reference.

  1. It would be helpful to label the groups (B16, OE, KO) in the figure more clearly, as the color bar at the top of the heatmap is not obvious. Additionally, it is better to keep the order consistency (AP1 vs others).

We thank the reviewer for this helpful observation. The heatmaps were generated in R using gene expression data, and the sample ordering reflects unsupervised clustering performed by the software. To preserve the integrity of this clustering and the biological structure it represents, we did not modify the sample arrangement. However, to enhance clarity, we have now explicitly labelled the B16, OE, and KO groups on the figure, ensuring that group identities are clearly visible without compromising the data presentation.

Round 2

Reviewer 2 Report

Comments and Suggestions for Authors

The authors addressed and explained all the comments I had raised in my previous review.